# Phytochemical Profiles, Antioxidant Activity and Antiproliferative Mechanism of *Rhodiola rosea* L. Phenolic Extract

**DOI:** 10.3390/nu14173602

**Published:** 2022-08-31

**Authors:** Sheng Zhang, Siqi Jiang, Na Deng, Bisheng Zheng, Tong Li, Rui Hai Liu

**Affiliations:** 1School of Food Science and Engineering, South China University of Technology, Guangzhou 510641, China; 2Institute of Human Nutrition, Columbia University Medical Center, New York, NY 10032, USA; 3School of Food Science and Bioengineering, Changsha University of Science & Technology, Changsha 410114, China; 4Overseas Expertise Introduction Center for Discipline Innovation of Food Nutrition and Human Health (111 Center), Guangzhou 510641, China; 5Guangdong ERA Food & Life Health Research Institute, Guangzhou 510670, China; 6Department of Food Science, Cornell University, Ithaca, NY 14853, USA

**Keywords:** *Rhodiola rosea*, phytochemical profiles, antiproliferative activity, cell apoptosis, cell cycle, p53 signaling pathway

## Abstract

The phenolic profiles, antioxidant activity, antiproliferative property and the underlying molecular mechanisms of cell apoptosis of *Rhodiola rosea* free phenolic (RFE) were analyzed in this work. Overall, *Rhodiola rosea* rhizome phenolic extract (RE) contained *Rhodiola rosea* rhizome free phenolic extract (RFE) and *Rhodiola rosea* rhizome bound phenolic extract (RBE). Compared with RBE, RFE contained higher phenolic contents and possessed stronger antioxidant activity. High-performance liquid chromatography (HPLC) results demonstrated that the main phenolics of were epigallocatechin (EGC), epigallocatechin gallate (EGCG), gallic acid (GA) and catechin. Gas chromatography–mass spectrometry (GC-MS) analysis found that *Rhodiola rosea* L. was rich in volatile phytochemicals. In addition, many types of vitamin E and a few kinds of carotenoids were found in *Rhodiola rosea*. In addition, the main compounds in RFE (GA, EGC, EGCG) and RFE all exhibited excellent antiproliferative activity, indicating the antiproliferative activity of RFE was partly attributed to the synergy effects of the main compounds. Further study confirmed that RFE could block 16.99% of HepG2 cells at S phase and induce 20.32% programmed cell death compared with the control group. Specifically, RFE dose-dependently induced cell apoptosis and cell cycle arrest via modulating the p53 signaling pathway including up-regulation of the expression of p53 and Bax while down-regulation of the Bcl-2, cyclin D1 and CDK4 levels. Therefore, RFE exhibited the potential of being developed as an auxiliary antioxidant and a therapeutic agent for cancer.

## 1. Introduction

Cancer triggers unacceptable high mortality. Currently, we are confronted with the obstacles of solving the problem of increased cancer incidence and mortality. Cancer may result from three main elements, including genetics, diet and environment [1]. Specifically, 35% of all cancers and 80% of colon cancer cases are caused by incorrect diet [1]. However, there are no extremely effective drugs for most cancers, and the existing treatments are accompanied by high cost and environmental hazard. Hence, it is necessary to find supplements which possess low side effects and environmentally friendly properties as suitable diets to manage the onset and development of cancers.

Previous research suggested that the health benefits of plants were correlated with the interaction or synergistic effects of bioactive compounds and other nutrients [2]. Additionally, phytochemicals can be supplemented as dietary therapy in cancer prophylaxis and treatment. Many of them have been proved to be effective as chemoprotective agents against commonly occurring cancers via interfering with tumor promotion and progression [3,4]. For instance, previous findings confirmed that dietary intake of carotenoids reduces the incidence of cancers [5]. Among all of the phytochemicals, phenolics have aroused considerable attention because of their various bioactivities, such as antioxidant, anti-inflammation, antiviral and antimicrobial [6]. For example, EGCG has been demonstrated to possess anticancer activity via various mechanisms, including inhibiting the growth of A549 cells via stabilization and activation of the p53 pathway as well as inhibiting the proliferation of HeLa cells by the induction of cell cycle arrest [7]. Except for that, gallic acid can trigger the PI3K/Akt pathway to inhibit the growth of A549 lung cancer cells [8]. Additionally, EGC exhibits the property of inducing the extrinsic and intrinsic pathways of apoptosis in neuroblastoma [9]. These functions may originate from their antioxidant, free radical scavenging and enzyme detoxifying regulation abilities due to the existence of hydroxyl groups and glycosylation or other substituents [3].

*Rhodiola* (*Rhodiola* L.), a member of the Crassulaceae family, is affiliated with perennial herbaceous plants [10], whose main distribution is the Qinghai-Tibetan Plateau and the adjacent areas. The abundant volatile compounds provide *Rhodiola* with a unique and strong fragrance. Moreover, the biological activities of volatile compounds in *Rhodiola* have been proved by previous reports. For example, myrtenol, as one of the main ingredients in volatile compounds of *Rhodiola* [11], showed anti-inflammatory and antioxidative activity in rats with allergic asthma [12]. *Geraniol*, another main volatile compound in *Rhodiola*, possessed various biological activities, such as anticancer, antimicrobial, antioxidant and some vascular effects [11,13]. In addition, *Rhodiola* is rich in various phytochemicals, including flavonoids, glycosides, phenolic compounds and organic acids [14]. Differing from rich tannins found in brown algae [15], the rhizome of *Rhodiola rosea* contained higher levels of phenolic acid and flavonoids. As a traditional Chinese herb, *Rhodiola* attracts a lot of attention because of its potential of being used as a therapeutic agent for improving the nervous system and preventing high-altitude sickness [14]. Concretely, numerous studies have focused on salidroside, the main content of *Rhodiola*. For instance, excellent antiproliferative activity of salidroside was found in MDA-MB-231 and A549 cells [16].

In conclusion, few studies on the exact contents of V_E_, carotenoids, volatile compounds and phenolics in *Rhodiola rosea* are available at present. Besides the function and synergistic effects of phenolics, other biologically active ingredients in *Rhodiola rosea* have rarely been investigated. Therefore, further explorations on the phenolics and bioactivities of *Rhodiola rosea* are urgently warranted. The primary aim of this work was to evaluate the phytochemicals of *Rhodiola rosea* and explore the antioxidant activity as well as potential antiproliferative mechanism of *Rhodiola rosea* free phenolic extract (RFE), which could further provide partial theoretical guidance for its use in cancer treatment.

## 2. Materials and Methods

### 2.1. Chemicals and Reagents

Fresh rhizomes of *Rhodiola rosea* L. were purchased from a local pharmacy in Yushu, Qinghai Province (33.00°, 97.02°), China, in 2019, and kept in a −20 °C refrigerator. The species was identified by Prof. Shizhen Ma (Northwest Institute of Plateau Biology, Chinese Academy of Sciences, Qinghai, China). L-ascorbic acid (ASA), α, β, γ, δ-tocopherol and α, β, γ, δ-tocotrienol were purchased from Wako Pure Chemical Industries (Tokyo, Japan) and Chromadex Ltd. (Irvine, CA, USA), respectively. Human liver cancer cell line HepG2 (ATCC HB-8065) were provided by ATCC (Manassas, VA, USA). Other chemicals and reagents were of analytical grade.

### 2.2. Extraction of Rhodiola Phenolics

#### 2.2.1. Extraction of RFE

Extraction of RFE was performed with the method mentioned before [17]. Briefly, the *Rhodiola rosea* L. rhizomes of suitable size and in good condition were crushed into powder, sieved by a 40-mesh sieve and stored in a desiccator. Thereafter, 2 g powder were extracted with 30 mL chilled 80% acetone, and the mixtures were homogenized (3 min at 12,000 rpm) and centrifugated (6000× *g* rpm for 3 min) to obtain supernatant. Following three duplications of the former steps, the supernatants were evaporated to dryness with a rotary evaporator at 55 °C and redissolved with 10 mL of distilled water. The acquired samples were kept in the refrigerator at −20 °C for further use.

#### 2.2.2. Extraction of RBE

The residual solids from RFE were utilized for the extraction of RBE [17]. In short, the residues and 4 mol/L NaOH solution were fully mixed at 60 rpm for 1.5 h, and then pH adjustment was carried out with 12 mol/L HCl and the ultimate pH was 2. Before conducting centrifugation (6000× *g* rpm for 5 min) to yield the top layer solution, the mixtures were extracted by 40 mL of ethyl acetate. After 5 duplications of the former procedures, the following steps for the acquirement of RBE were the same as in Section 2.2.1.

### 2.3. Determination of Total Polyphenols

A Folin–Ciocalteu colorimetric approach was used to determine the total polyphenol content [18]. Final results were expressed as milligrams of gallic acid equivalents per gram fresh weight of *Rhodiola* (mg GAE/g FW).

### 2.4. Analysis of Phenolic Profiles in Rhodiola rosea by HPLC

The identification of the polyphenol composition of *Rhodiola rosea* was performed by high-performance liquid chromatography (HPLC) (Waters Corporation, Milford, MA, USA). In brief, the gradient of binary elution phase (A: acetonitrile, B: 0.1% formic acid in Milli-Q water) was as follows: 0–5 min (8% A), 5–10 min (8–15% A), 10–25 min (15–35% A), 25–30 min (35–50% A), 30–35 min (50–100% A), 35–36 min (100% A–5% A), 36–50 min (5% A) at a flow rate of 0.8 mL/min. The external standard method was used to detect the polyphenol compounds in *Rhodiola rosea*. Results were presented as milligrams per 100 g fresh weight of *Rhodiola* (mg/100 g FW, mean ± SD, *n* = 3).

### 2.5. Profiles of Volatile Compounds in Rhodiola rosea Detected by Static Headspace-Gas Chromatography–Tandem Mass Spectrometry (GC-MS)

The identification of volatile compounds in *Rhodiola rosea* was performed by an Agilent 7890A-7000C (Agilent Technologies, Palo Alto, CA, USA). The static headspace instrumentation is composed of an Agilent system, a headspace auto sampler, a heater and an agitator. The sample was heated to 45 °C and incubated for 15 min with the agitator rotating at 500 rpm. The interval time of shaking was 2 s and each shaking process was required to last for 5 s. The sample extraction time and the desorption time at the gas injection were 45 min and 4 min, respectively. Finally, the headspace gas was injected with a split ratio of 50:1. The GC-MS analysis was performed by an Agilent 7000C equipped with HPINNOWAX (30 mm × 0.25 mm, 0.25 μm) and all mass spectra were acquired in electron impact ionization (EI) mode. A mass spectra database search (NIST14) was used to identify the volatile compounds in samples and the relative quantitative analysis was performed by peak area measurement.

### 2.6. Extraction and Profiles of Vitamin E and Carotenoids

#### 2.6.1. Extraction of Vitamin E and Carotenoids

Vitamin E (V_E_) was extracted with the method mentioned before with some modification [19]. Briefly, the powder of *Rhodiola* was mixed with 95% ethanol solution containing pyrogallol, sodium chloride and ascorbic acid. In the following step, KOH (600 g L^−1^) was added into the mixture and n-hexane/ethyl acetate (9:1, *v*/*v*) was used for extraction. Finally, the organic layer was collected and dried under nitrogen. Residues were reconstituted with n-hexane solution (containing 1% isopropyl alcohol) for vitamin E analysis and MTBE solution (containing 1% BHT) for carotenoid analysis, separately.

#### 2.6.2. Vitamin E Analysis by NP-HPLC

The analysis of vitamin E was carried out with the HPLC method described before [19] coupled with an Agilent ZORBAX RX-SIL column (250 mm × 4.6 mm, 5 μm) and the Breeze system. The mobile phase contained 0.85% isopropyl alcohol in n-hexane (*v*/*v*) and 0.1% acetic acid with the flow rate of 1 mL/min. The excitation wavelength and the emission wavelength were 290 nm and 330 nm, respectively. Data were expressed as micrograms per gram of sample fresh weight (μg/g FW) (mean ± SD, *n* = 3).

#### 2.6.3. Carotenoid Analysis by HPLC

The analysis process of carotenoids was performed by an HPLC system (Waters Corporation, Milford, MA, USA) equipped with a YMC carotenoid 30 column (4.5 × 250 mm, 5 μm) and a photodiode array detector with the previously reported method [19]. Mobile phase A was composed of 0.1% (*w*/*v*) BHT and 0.05 M ammonium acetate in 97% (*w*/*v*) methanol–water and mobile phase B consisted of 0.1% (*w*/*v*) BHT in methyl tert-butyl ether. The gradient elution was as follows: 0–18 min 0–20%B; 18–20 min, 20–50% B; 20–25 min, 50–90% B; 29–29.5 min, 90–10% B; and 29.5–40 min, 10–0% B. The data were expressed as micrograms per gram of sample fresh weight (μg/g FW) and were reported as mean ± SD in triplicate.

### 2.7. Cell Culture

HepG2 human liver cancer cells was cultured in DMEM supplemented with 10% FBS and 1% antibiotic antimycotic solution and maintained in a 5% CO_2_ incubator at 37 °C. All cell culture reagents were purchased from Gibco (Life Technologies, Grand Island, NY, USA).

### 2.8. Antioxidant Activity Assay

#### 2.8.1. In Vitro Antioxidant Assay

The oxygen radical absorbance capacity (ORAC) and peroxyl radical-scavenging capacity (PSC) assays were utilized to evaluate the in vitro antioxidant activity of *Rhodiola rosea* phenolic extract (RE), referring to the method described before [20]. PSC and ORAC values were expressed as μmol vitamin C equivalent per gram dry weight (μmol VCE/g DW) and μmol Trolox equivalents per gram dry weight (μmol TE/g DW), separately (mean ± SD, *n* = 3).

#### 2.8.2. Cellular Antioxidant Activity (CAA) Assay

CAA assay was performed to further assess the antioxidant capacity of RE, using the methods reported previously [20]. Final values were presented as µmol quercetin equivalent per gram of dry weight of *Rhodiola* (µmol QE/g DW).

### 2.9. Cytotoxicity and Antiproliferative Activity Assays

Methylene blue assay was performed to test the antiproliferative effect and cytotoxicity of RFE towards HepG2 human cancer cells [20]. In terms of the antiproliferative assay, 1.5 × 10^5^ cells/mL were seeded in each well, followed by incubating the cells with RFE for 72 h. For cytotoxicity test, the initial density of cells and incubation time were 2.5 × 10^5^ cells/mL and 24 h, respectively. The effects of RFE on antiproliferative activity and cytotoxicity were estimated with the median effective concentration (EC_50_) and half-maximal cytotoxicity concentration (CC_50_), separately. The concentrations triggering less than 20% cells to die were regarded as non-toxic.

### 2.10. The Determination of Synergy Effects

In order to investigate if individual phenolics had a synergistic effect on the antiproliferative activity of RFE, the effect of the combination of the main monomers (GA, EGC and EGCG) on HepG2 cells was determined by synergy determination assay as mentioned before [21]. Based on the EC_50_ of the antiproliferative effects of GA, EGC and EGCG, the concentrations for synergistic effect determination of any two of the monomers were set as follows: 0.125 × EC_50_, 0.25 × EC_50_, 0.5 × EC_50_, 0.75 × EC_50_, 1.0 × EC_50_ and 1.25 × EC_50_. The synergy concentrations of the three monomers were set as follows: the concentration of the two mixtures was multiplied by 2/3. The synergy indexes were calculated with the following equations:(1)$$\mathrm{CI}=\frac{{\left(D\right)}_{1}}{{\left(D_{X}\right)}_{1}}+\frac{{\left(D\right)}_{2}}{{\left(D_{X}\right)}_{2}},$$
(2)$${\left(\mathrm{DRI}\right)}_{1}=\frac{{\left(D_{X}\right)}_{1}}{{\left(D\right)}_{1}},\;{\left(\mathrm{DRI}\right)}_{2}=\frac{{\left(D_{X}\right)}_{2}}{{\left(D\right)}_{2}},$$
where (D)_1_ and (D)_2_ are the single drug concentrations when the two drugs acted in combination, respectively, and (D)_x_ is the respective concentration when the two drugs acted alone to achieve the same inhibition rate. CI < 1, CI = 1 and CI > 1 represent synergistic, additive and antagonistic effects between the drugs, respectively. (DRI)_1_ and (DRI)_2_ represent the dose reduction index of each drug under interaction. The CI and DRI values were calculated with CalcuSyn software version 2.0 (Biosoft, Cambridge, UK).

### 2.11. Cell Cycle and Cell Apoptosis Detection Assay

HepG2 cells (2.5 × 10^5^ cells/mL) were cultured in six-well microplates for 12 h, and then cells were treated with medium containing (or not containing) different doses of RFE for another 36 h. Thereafter, all of the cells, including floating cells, were collected, and the apoptotic effect of RFE on HepG2 cells was determined by the corresponding assay kits (Nanjing Jiancheng Bioengineering Institute, Nanjing, China). Cell cycle arrest was analyzed by Modifit software (Phoenix Flow Systems, Inc., San Diego, CA, USA).

### 2.12. RT-qPCR Analysis

After incubating in the same conditions depicted in Section 2.11, the extraction for total RNA of HepG2 cells was performed by TRIzol reagent. Then, the AG Evo M-MLV Mix Kit with gDNA Clean for qPCR (Accurate Biology, Hunan, China) was used to reverse the RNA into cDNA. Based on the previously reported method [20], the RT-qPCR was executed under the conditions of denaturation at 95 °C for 10 s, followed by 40 cycles of amplification (95 °C for 5 s, and 60 °C for 30 s). The primers used in RT-PCR are depicted in Appendix A. The data analysis was performed by the 2^−ΔΔCt^ method, and results were expressed as relative expression compared with the control group.

### 2.13. Western Blot Analysis

Cell collection and lysis: Cells were collected using a cell scraper after treatment under the same conditions described in Section 2.11. Then, 200 μL of RIPA cell lysate (containing 10 mM of PMSF) was added into each group. Thereafter, the protein contents were determined using the BCA method.

Cellular protein processing: The samples were mixed with loading buffer in an appropriate ratio and centrifuged. The mixed samples were placed in boiling water (10 min) for protein denaturation.

SDS-PAGE electrophoresis: Gel preparation was performed in a gel-making chuck and the appropriate gel concentration was selected according to the protein molecular weight according to the instructions of the Beyond Rapid Gel Preparation Kit (P0012AC). After adding electrophoresis buffer into the electrophoresis tank, samples and pre-stained marker were added into each well, and electrophoresis was carried out at the voltage of 60 V and sustained for 30 min. When the proteins ran close to the separation gel, the voltage was increased to 100 V, continuing electrophoresis for another 100 min and stopping electrophoresis when all samples ran to the bottom of the separation gel.

Transfer to membrane: The gel was transferred from the glass plate into transfer buffer in advance. PVDF membrane (pre-activated by methanol), gel and another two layers of filter paper were placed suitably to form a sandwich structure of “cathode–2 layers of filter papers–gel-membrane–2 layers of filter papers”. Afterwards, the sandwich structure was installed with a slot. The transfer process lasted for 45 min with the voltage of 100 V in an ice bath.

Blocking: At the end of membrane transfer step, the PVDF membrane was rinsed with TBST and incubated with 5% fat-free milk powder solution (diluted by TBST) for 2 h at room temperature.

Primary antibody incubation: After the blocking step was completed, the 5% fat-free milk powder solution was replaced by 15 mL of TBST solution for rinsing the membrane, repeating 3 times for 5 min each time. After that, the membrane was incubated with diluted primary antibody (p53, Bcl-2, Bax, CDK4, cyclin D1) overnight at 4 °C.

Secondary antibody incubation: TBST solution was added to wash the membrane in triplicate for 10 min each time on a shaker. There should be no air bubbles at the bottom of the membrane after the ultimate wash, and then secondary antibody (diluted with TBST at a ratio of 1:3000) was added into the incubation cassette, incubating with the membrane for 2 h at room temperature.

Chemiluminescence: After washing the membrane with the same steps as before, the prepared chromogenic solution (A:B = 1:1) was evenly sprinkled on the membrane. Then, chemiluminescence was immediately performed and the final signal was exhibited on X-ray film using an X-dark box.

### 2.14. Statistical Analysis

Data were reported as mean ± standard deviation (SD) with triplicates. Statistical analyses were performed by IBM SPSS 23.0 (SPSS Inc., Chicago, IL, USA), using one-way ANOVA to determine statistical significance. *p* < 0.05 was the standard to consider differences as significant. The EC_50_ and CC_50_ values were calculated by CalcuSyn software version 2.0 (Biosoft, Cambridge, UK).

## 3. Results and Discussion

### 3.1. Analysis of Phenolic Contents in RFE

As shown in Table 1, 120.13 ± 2.88 mg GAE/g FW of free polyphenol and 125.71 ± 6.06 mg CE/g FW of free flavonoid were observed in *Rhodiola rosea*, whereas the contents of bound polyphenol and bound flavonoid were 2.58 ± 0.22 mg GAE/g FW and 3.28 ± 0.24 mg CE/g FW, respectively. In the case of the low contents of bound polyphenol, RFE in *Rhodiola rosea* was considered as the main research object. Table 1 and Figure 1 show that the main ingredient in RFE was EGCG (722.10 ± 54.26 mg/100 g FW), which was followed by gallic acid (319.17 ± 41.18 mg/100 g FW). The EGC ranked third with the content of 267.04 ± 16.09 mg/100 g FW, followed by catechin (35.93 ± 7.39 mg/100 g FW).

As EGCG is an indicator of antioxidant activity [22], *Rhodiola* is expected to have great potential in antioxidant ability. Polyphenol compounds attract much attention because of their outstanding radical-scavenging properties to prevent chronic and oxidative stress-related ailments. To summarize, *Rhodiola* exhibits the potential to be utilized as a promising antioxidant and remedy of some ailments.

### 3.2. Profiles of Volatile Compounds in Rhodiola rosea

GC-MS analysis was efficient to identify the exact compounds of volatiles. As shown in Table 2, *Rhodiola rosea* consisted of several complex compounds. In *Rhodiola rosea*, the richest volatile was phenylethyl alcohol, accounting for 7869.33 ± 174.51 ng/100 g FW, followed by (-)-myrtenol (1477.80 ± 54.76 ng/100 g FW) and 1-octanol (848.14 ± 19.60 ng/100 g FW). Geraniol and acetic acid shared similar contents with values of 550.87 ± 21.27 and 514.48 ± 6.77 ng/100 g FW, respectively.

The difference in volatile components in *Rhodiola* might explain the unique flavor of samples. In addition, these dominant compounds, containing hydroxyl groups, tended to be polar molecules that exhibited special physiological activities. For instance, phenylethyl alcohol exhibited a sedative effect on mice [23]. Additionally, geraniol could suppress the proliferation of hepatocellular carcinoma [24]. Mariusz Kwon et al. [25] also proved the antitumor activity of myrtenol on colon carcinoma cells. In addition, geraniol and myrtenol have been confirmed to be effective in exerting excellent antioxidant activity [26,27]. Hence, we could conclude that the excellent antioxidant and antiproliferative activity of *Rhodiola* might be correlated with its rich volatile compounds, which implied its promising future of exploitation as a functional species.

### 3.3. Analysis of V_E_ and Carotenoid Contents in Rhodiola rosea

The tocochromanol and carotenoid profiles in *Rhodiola rosea* were identified by the HPLC method, and data are listed in Table 3. Overall, α-tocopherol, α-tocotrienol, β-tocotrienol, γ-tocopherols, γ-tocotrienol and δ-tocotrienol were detected in RFE. Among the six variations in vitamin E, the content of α-tocopherol (accounting for 338.51 ± 17.75 μg per 100 g FW) ranked first which was followed by β-tocotrienol, γ-tocotrienol, δ-tocotrienol and γ-tocopherols, while α-tocotrienol was the lowest (16.82 ± 0.47 μg per 100 g FW). For carotenoids, the total content of carotenoids in *Rhodiola rosea* was 7.43 ± 0.72 μg per 100 g FW. There were two main carotenoids existing in *Rhodiola rosea*, namely lutein and β-carotene, accounting for 3.48 ± 0.11 μg per 100 g FW and 3.95 ± 0.40 μg per 100 g FW, respectively.

Evidently, *Rhodiola* shared many similar positive effects with V_E_, such as antiaging, antioxidant, preventing cardiovascular diseases and improving sperm quality [14,28,29]. However, there have been few articles on the contents of V_E_ in *Rhodiola*. Hence, the similarity and scarcity attracted our attention to the exact profiles of V_E_ in *Rhodiola*. In addition, carotenoids, widely distributed in plants and animals, are fundamental natural pigments [30]. Similarly, *Rhodiola* exhibited analogous features to carotenoids, including protecting eyes from some disorders, anti-inflammation and treating some types of cancers [14,31,32]. Thus, the determination of carotenoid profiles in *Rhodiola* is critical to evaluate the functional diversity of *Rhodiola*.

### 3.4. In Vitro Antioxidant Activity of Rhodiola rosea Phenolics

As presented in Figure 2A, the total PSC and ORAC values of RE were 353.78 ± 15.75 μM VCE/g DW and 1807.40 ± 45.63 μM TE/g DW, respectively. The PSC and ORAC values of RFE (342.22 ± 15.83 μM VCE/g DW and 1644.06 ± 46.50 μM TE/g DW, respectively) were close to the corresponding data of total RE which were over 28 times and 9 times of that of RBE, respectively. This phenomenon implied that RFE played a crucial role in the excellent antioxidant ability of RE. As the main polyphenols in *Rhodiola rosea*, the PSC and ORAC values of EGCG were 11.52 ± 1.27 μM VCE/g DW and 14.62 ± 1.33 μM TE/g DW, respectively.

A previous study reported that the strong antioxidant property of some plants could be attributed to the high contents of phenolics [33]. Additionally, the free radical scavenging ability of samples could be detected by PSC and ORAC assays. Hence, it can be concluded from Figure 2A that RFE with richer phenolics was more efficient than RBE in radical-scavenging ability. The PSC and ORAC values suggested the incomparable effect of *Rhodiola rosea* phenolics in antioxidant ability which were more than 4 times and 12 times of that of red kiwifruit (85.96 ± 11.75 μM vitamin C equivalent/g FW and 131.23 ± 5.91 μM Trolox equivalent/g FW, respectively) [34]. These results indicated that *Rhodiola rosea* free phenolics have the potential for being an auxiliary agent to resist oxidative damage and treat related diseases.

### 3.5. Cellular Antioxidant Activity of RE

Traditional in vitro chemical antioxidant assays disregard the absorption and metabolism of polyphenols in vivo. Therefore, the determination of the antioxidant activity of RE was performed by CAA assay. Compared with the traditional ways, the CAA assay is unparalleled and excellent because of many advantages, including simulating the process of cellular metabolism as well as forecasting the antioxidant behavior in cells. CAA quality was used to measure the cellular availability of polyphenols or flavonoids. Moreover, the percentage of CAA values with PBS wash and no PBS wash was calculated to quantify the exact extent of cellular absorption of polyphenol. As depicted in Figure 2B, the CAA value of RFE with PBS wash was 77.29 ± 4.95 μmol QE/g DW, which was approximately 70% of the total CAA value in the no PBS wash protocol (108.19 ± 0.66 μmol QE/g DW). The total CAA values with (78.05 ± 4.96 μmol QE/g DW) or without PBS wash (110.34 ± 0.77 μmol QE/g DW) were close to the CAA values of RFE. Compared with CAA values of RFE, the CAA values of RBE were too insignificant (0.76 ± 0.01 μmol QE/g DW in PBS wash while 2.19 ± 0.09 μmol QE/g DW in no PBS wash). EGCG, as the richest compound in RFE, did not show excellent cellular antioxidant ability and the CAA values of EGCG with and without PBS wash were 5.20 ± 0.29 μmol QE/g DW and 11.62 ± 0.58 μmol QE/g DW, respectively, implying the weak correlation between the cellular antioxidant ability of RFE and the content of EGCG. Furthermore, as the data show in Table 4**,** RFE showed far better CAA quality than traditional fruits and vegetables, including some “super fruits” [35]. As for PBS wash, the CAA quality of *Rhodiola rosea* phenolics was more than 5 times that of blueberry and 10 times that of green grape, while the difference in no PBS wash shared a similar trend (more than 1.5 times that of blueberry and 14 times that of green grape, respectively) [36]. In addition, the CAA value of RFE (no PBS wash) was more than 1200 times that of the data of Jingu34 foxtail millet and 1800-fold that of M2504-6 Proso millet, suggesting the remarkable antioxidant activity of RFE at the cellular level.

### 3.6. Cytotoxicity and Antiproliferative Effects of RFE

HepG2 cells were utilized to establish the antiproliferative model as well as to determine the cytotoxic and antiproliferative activity of RFE, GA, EGC and EGCG. Overall, RFE and its three phenolic monomers showed no significant cytotoxicity in the concentration range (Figure 3). As shown in Figure 3, RFE and the three major monomers were effective in inhibiting the growth of HepG2 cells in a significant dose-dependent manner. Among the three major phenolic monomers of RFE, EGCG had the strongest antiproliferative activity, while EGC showed relatively weak antiproliferative activity. Antiproliferative activities of the samples were assessed by EC_50_, with a lower EC_50_ implying better antiproliferative activity. As illustrated in Table 5, the EC_50_ values of RFE and its three major monomers were 276.32 ± 8.47 μg/mL (RFE) and 73.91 ± 3.23 μM (GA), 107.21 ± 4.64 μM (EGC) and 61.66 ± 7.97 μM (EGCG), respectively. As a criterion to judge the antiproliferative and cytotoxic properties of the samples, selectivity index (SI) > 2 indicates the antiproliferative properties of the samples, while SI < 2 reflects the cytotoxicity of the samples, and SI is calculated as CC_50_/EC_50_. The SI values of RFE and its three phenolic monomers were all above 2 (Table 5), implying their potential antiproliferative ability. Phenolic compounds widely exist in natural plants. Except for investigating their antioxidant activity, many research results also reported their anticancer activity. Veeriah et al. [37] confirmed that an antiproliferative effect of phenolic compounds from apple was exerted on HT human colon cells. Therefore, we propose that the antiproliferative activity of *Rhodiola rosea* might be closely correlated with the rich contents of phenolic compounds. Furthermore, EGCG, the most abundant ingredient in the RFE, has been proved to be the most effective phenolic in inhibiting the proliferation of HCT-116 and SW-480 cells compared to other representative polyphenols in *Rhodiola rosea*, such as caffeic acid, gallic acid, catechin, epicatechin, gallocatechin, gallocatechin gallate and epicatechin gallate [38]. Therefore, a high content of EGCG might clarify the excellent antiproliferative effect of RFE, thus suggesting *Rhodiola rosea* as a promising antiproliferative agent in the future. However, in-depth research on the antiproliferative effects and mechanisms of individual phenolics in *Rhodiola rosea* is urgently required. Therefore, a series of assays were executed to determine the underlying mechanism of the extraordinary antiproliferative activity of RFE.

### 3.7. Combined Effect of GA, EGC and EGCG on Inhibiting the Proliferation of HepG2 Cells

To further explore the potential mechanism of antiproliferative activity in RFE, we investigated the interaction among the three main phenolic monomers (GA, EGC and EGCG) in *Rhodiola rosea* regarding the inhibition of growth of HepG2 cells. On the whole, GA, EGC and EGCG could interact with each other as combinations of two or three monomers. As shown in Table 6, the proliferation of HepG2 cells was dose-dependently inhibited either in two or three combinations. At lower concentrations (0.125–0.25 EC_50_), the antiproliferative effect of the GA + EGC combination group (18.97 ± 3.28%–32.29 ± 7.09%) was weaker than that of the GA + EGCG group (48.74 ± 3.18%–47.97 ± 9.53%) and EGC + EGCG group (33.81 ± 2.76%–54.22 ± 5.66%). In addition, the synergy of the three phenolic monomers showed a comparatively weaker antiproliferation effect on HepG2 cells (47.22 ± 3.89%–67.89 ± 0.59%) compared with the combination groups of two monomers at higher concentrations (0.50–1.25 EC_50_). However, the variability of inhibition effects on tumor cells existed at different doses of combination groups. Therefore, only considering the inhibition rates of tumor cells at low or high doses cannot accurately reflect the synergy effect of the main phenolic monomers in RFE, and thus combination index (CI) and dose reduction index (DRI) values were further analyzed to evaluate the combination effect of the phenolic monomers.

### 3.8. Analysis of CI and DRI Values

The CI value is an important index to indicate whether there is a synergy effect between different compounds, while the DRI value indicates the dose reduction index of each single drug when a synergy effect exists among these compounds. As shown in Figure 4A, the CI range of the GA + EGC group was 0.57–1.01 when the inhibition rate of HepG2 cells was 20–60%, indicating a synergistic effect between these two compounds. After the inhibition rate gradually increased to 80%, the corresponding CI value increased to 1.43, suggesting that GA and EGC exhibited an antagonistic effect in the GA + EGC group. Similarly, when the CI values in the GA + EGCG group (0.10 ± 0.01–0.90 ± 0.01), EGC + EGCG group (0.26 ± 0.20–0.92 ± 0.25) and GA + EGC + EGCG group (0.32 ± 0.08–0.97 ± 0.11) were smaller than 1, the inhibition rates of HepG2 cells were 20–65%, 20–70% and 20–50%, respectively, and then all of the four groups showed antagonistic effects with the increment in combined doses of these drugs. As one of the standards of the synergistic effects of drugs, the DRI values for each individual phenolic in various groups (0.2 ≤ Fa ≤ 0.8) are depicted in Figure 4B and Table 7. With a similar trend to CI values, the DRI values of GA (34.48 ± 5.51) and EGCG (14.24 ± 0.95) in the GA + EGCG combination group were markedly higher than the figures in the other groups when the inhibition rate of HepG2 cells was low (Fa = 0.2). When the inhibition rate of HepG2 cells reached 80%, the DRI value of GA decreased to 0.72 ± 0.12, which was consistent with the trend in the CI value. Except for the DRI values of GA in the GA + EGC + EGCG group (0.72 ± 0.09–0.97 ± 0.07) and in the GA + EGC group (0.72 ± 0.12), the DRI values of other groups were all larger than 1 when Fa was relatively larger (0.75 ≤ Fa ≤ 0.8), indicating that the interaction among phenolic compounds could reduce the doses of drugs. For screening antitumor drugs, Chou et al. [39] reported that CI value was more meaningful when the inhibition rate of tumor cells was close to 80%. In our study, when the inhibition rate of HepG2 cells was greater than 70%, the CI value of each group was larger than 1, which no longer showed a synergistic effect, but the DRI value indicated that the combination of drugs still had a guiding meaning for the diminution of drug doses. Altogether, the drug doses can be reduced appropriately by the combination of individual phenolics, thus the damage caused by potential toxic side effects of high-dose drugs to the human body can be alleviated without sacrificing their antitumor activity.

### 3.9. PCA Analysis

Principal component analysis (PCA) among the phenolic monomer combination groups was performed and is shown in Figure 5. Overall, 91.6% of variation among tested samples was attributed to PC1 (78.2%) and PC2 (13.4%). PC1 separation was mostly caused by DRI (EC_50_), CI (EC_50_), EC_50_, whereas PC2 was separated owing to CI (EC_50_) and EC_50_. Moreover, two monomers (GA and EGCG) in the GA + EGCG group possessed comparatively lower EC_50_ values, suggesting the relatively weaker antiproliferative activities of those groups. In addition, the GA + EGCG group showed comparatively larger DRI and smaller CI, suggesting the extraordinary synergistic effects on inhibiting the growth of HepG2 cells. Meanwhile, GA in the GA + EGC group and EGCG in the GA + EGC + EGCG group were close at PC1 levels, which had relatively larger CI and lower DRI values. This phenomenon implied the fact that they were similar in some features, such as poor synergistic effects and inhibitory effects.

### 3.10. Effect of RFE on Cell Cycle Arrest

G_0_/G_1_ phase, S phase and G_2_/M phase constitute the complete cell cycle. Different DNA contents exist in different cell stages, thus the potential antiproliferative mechanism can be evaluated by comparing the change in DNA contents among various groups. The concentrations of RFE (240 μg/mL, 360 μg/mL and 480 μg/mL) were selected based on the EC_50_ values of RFE against the proliferation of HepG2 cells (Table 5). After supplementing with RFE, 29.76–46.75% of HepG2 cells were dose-dependently arrested in the S phase (Figure 6), confirming the inhibitory effect of RFE on the proliferation of HepG2 cells. After intervention with RFE, a decrease from 68.72% to 51.90% of HepG2 cells in the G1 phase could be observed, while the proportion of cells in the G2 phase fluctuated up and down slightly.

### 3.11. Effect of RFE on Apoptosis Induction

The effect of RFE on cell apoptosis was evaluated and is depicted in Figure 7. The apoptosis rate of HepG2 cells was dose-dependently enhanced after exposure to RFE. An increase from 11% to 31.32% in the proportion of apoptotic cells could be found with the increased doses of RFE. After administration with 240 and 360 μg/mL RFE, most HepG2 cells exhibited early apoptosis, where the early apoptosis rate of cells was more than twice the late apoptosis rate. However, the percentages of early and late apoptotic cells (12.08% and 19.24%, respectively) were similar to each other when the concentration of RFE was 480 μg/mL.

The resistance to cell apoptosis is one of the defining hallmarks of cancer. Thus, there is a necessity of finding an agent that can boost the apoptosis of cancer cells. As mentioned before, EGCG has been demonstrated to be the main phenolic in *Rhodiola rosea* to promote the apoptosis of cancer cells. Consequently, it is reasonable to attribute the cause of cell apoptosis to EGCG in this work. The present study showed that RFE could induce the apoptosis of HepG2 cells, but further exploration of the potential molecular mechanisms of the phenomenon is needed.

### 3.12. Effect of RFE on Mitochondrial Pathway-Related Gene and Protein Expression

To further investigate the potential antiproliferative mechanism of RFE, the expression of proliferation-related genes was examined by RT-PCR. As illustrated in Figure 8A, in contrast with the control group, the expressions of *cyclin D1*, *CDK4*, *Bcl-2* were dose-dependently down-regulated, while the levels of *Bax*, *p53* were up-regulated dose-dependently after supplementation with RFE. Concretely, high-dose RFE treatment (480 μg/mL) significantly augmented the relative expression levels of *p53* and *Bax* by 4.54 and 6.89 times, respectively, while that of *Bcl-2* was down-regulated by more than 16 times compared to the control group. Cyclin and CDKs proteins are the key proteins to adjust cell cycle, and the cyclin–CDKs complex is promising to regulate cell metabolism, thereby inhibiting the proliferation of cells. Accordingly, the down-regulation trends of the expression levels of *cyclin D1* and *CDK4* were positively correlated with the RFE concentrations, and their greatest reductions were 0.54 and 0.27 times, respectively. Since the strongest antiproliferative activity was observed in the high-dose RFE treatment group, subsequent Western blotting analysis was executed based on this concentration (480 μg/mL). Our results found that the regulation of cyclin D1, CDK4, Bcl-2, Bax and p53 protein levels showed a similar trend to the corresponding genes (Figure 8B,C). Following 36 h of treatment with RFE, the down-regulation trend could be observed in the relative expression levels of Bcl-2 and CDK4 proteins compared to the control group (declined by 0.28 times), and the largest down-regulation level was that of cyclin D1 protein (0.36 times). By contrast, the relative expression of Bax and p53 was up-regulated to 2.54 and 1.64 times that of the control group, respectively.

p53 regulates the expression of genes involved in various cellular activities, including cell apoptosis and cell cycle arrest [40]. CDKs, regulated by p53, can regulate the cell cycle. For instance, the cyclin A/CDK2 complex regulates the transition of S phase [41]. Correspondingly, the expressions of cyclin A and CDK2 exhibited upward trends, which were consistent with the result of the cell cycle arrested in the S phase. Additionally, CDK2 is the regulator of the G1–S transition [42]. Moreover, the formation of the cyclin D-CDK4/6 complex also engages in the transition of G1 to S phase [43]. Therefore, the inhibition of the activity of CDK4/6 can be the potential reason for arrested cell cycle of HepG2 cells in the G1 phase. Closely agreeing with the changes in the relative expression of CDK2, cyclin D1 and CDK4 proteins, cell cycle assay showed that the percentage of cells in the G1 phase decreased from 68.72% to 51.90%, while the figure for S phase increased from 29.76% to 46.75%, implying the phase transfer of cell cycle in HepG2 cells from G1 phase to S phase.

In the previous reports, p53 is regarded as an apoptotic gene [44]. For instance, the apoptotic process of HepG2 cells caused by eurycomanone involves the process of up-regulating p53 and Bax levels as well as down-regulating Bcl-2 expression [45], which was in alignment with our research. Moreover, a close link exists between p53 and Bcl-2 family members. For example, Bcl-2 family members could be the potential mediators of p53 during the process of cell apoptosis.

Mitochondrial outer membrane permeabilization (MOMP) is often observed in cells after being exposed to apoptosis stimulus. MOMP, controlled by Bcl-2 family proteins, can accelerate the process of cell apoptosis or counteract the process [46]. By balancing antiapoptotic and proapoptotic proteins, the final signal promoting survival or death is determined. For example, Bax plays a pivotal role in the process of MOMP. Moreover, BAD, another proapoptotic protein, accelerates this process via regulating Bcl-2 family proteins, including up-regulation of Bax and down-regulation of Bcl-2. Afterwards, cytochrome C tends to be released from mitochondria, and forms apoptosomes to promote the expression of caspase-9, triggering other caspase proteins, such as caspase-3. Eventually, the induction of caspase-3 can lead to cell apoptosis [47]. Thus, we propose that RFE possessed excellent antiproliferative activity via the p53 signaling pathway, thereby including cell cycle arrest and mitochondrial pathways.

## 4. Conclusions

The present study showed that *Rhodiola rosea* free phenolic extract was rich in phenolics and flavonoids. Among all the detected phenolic compounds, the content of EGCG was the highest, followed by gallic acid, EGC and catechin. Moreover, other phytochemical profiles, including V_E_, carotenoids and volatile compounds, in *Rhodiola rosea* were determined in this work. Furthermore, the favorable antioxidant activity of RFE was confirmed by CAA, PSC and ORAC assays. RFE also showed remarkable antiproliferative activity against HepG2 cells, which was correlated with the synergy effects of the main phenolic compounds in RFE. The antiproliferative mechanism of RFE was found to be attributed to ability of promoting cell apoptosis, arresting cell cycle in S phase as well as modulating the p53 signaling pathway via down-regulation of Bcl-2, cyclin D1 and CDK4 levels and up-regulation of p53 and Bax. In summary, the results illustrated the potential of RFE to be an excellent antioxidant and antiproliferative supplement.

## Figures and Tables

**Figure 1 nutrients-14-03602-f001:**
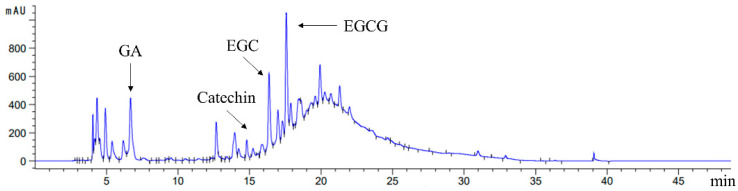
HPLC chromatogram of phenolic profiles in *Rhodiola rosea* free phenolic extract (RFE) at 285 nm.

**Figure 2 nutrients-14-03602-f002:**
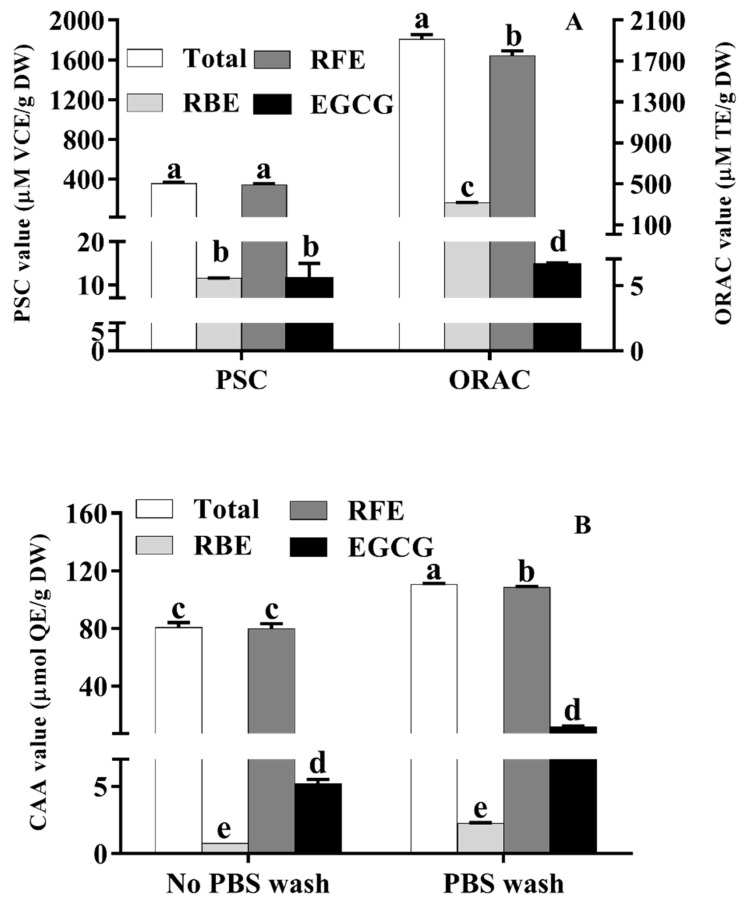
The PSC, ORAC (**A**) and CAA values (**B**) of RFE, RBE and EGCG (mean ± SD, *n* = 3). Bars with different letters differ significantly at *p* < 0.05.

**Figure 3 nutrients-14-03602-f003:**
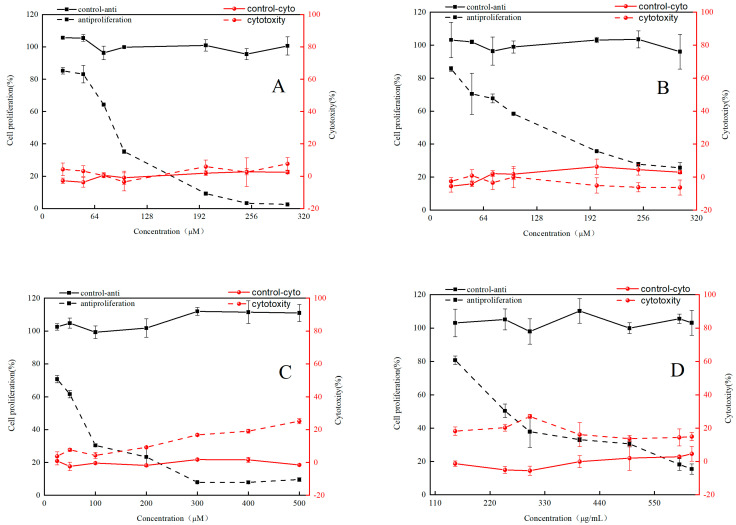
The cytotoxicity and antiproliferative activity of RFE and GA, EGC and EGCG on HepG2 human liver cancer cells: RFE (**A**); EGC (**B**); EGCG (**C**); RFE (**D**) (mean ± SD, *n* = 3).

**Figure 4 nutrients-14-03602-f004:**
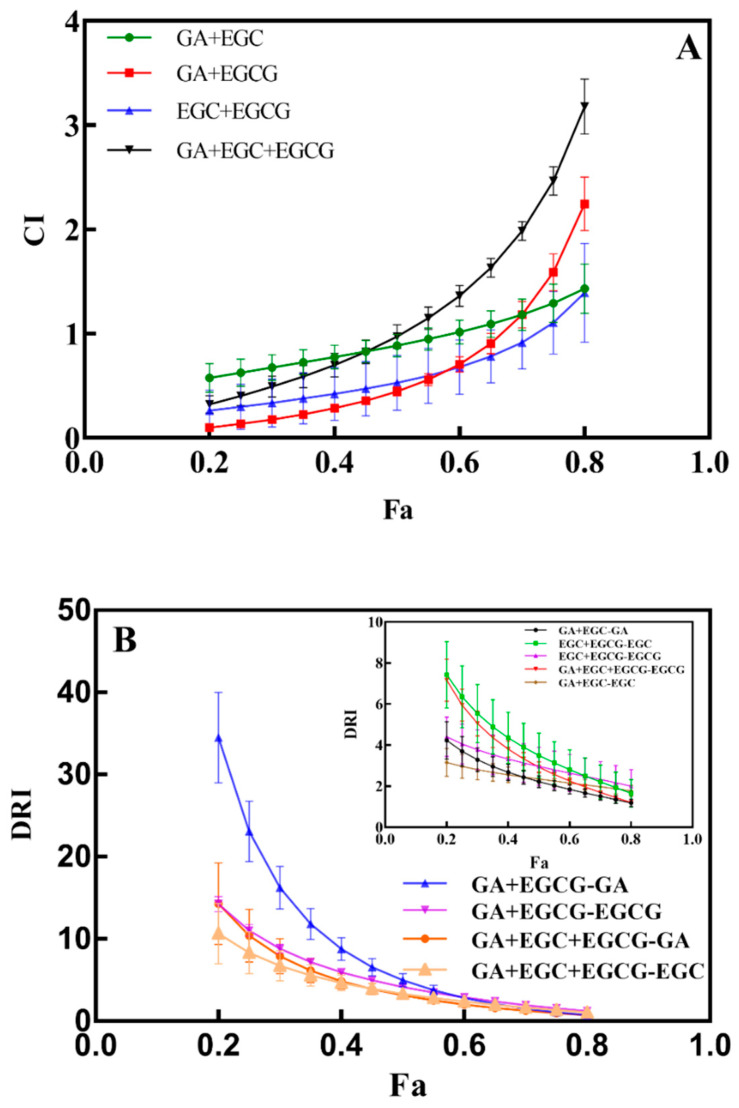
The correlations between inhibitory effects (Fa) and CI value (**A**), as well as Fa and DRI value (**B**) in HepG2 cells. GA + EGC, GA + EGCG, GA + EGC + EGCG in Figure (**A**) represent combined drug groups of GA and EGC, GA and EGCG, GA, EGC and EGCG, respectively; while GA + EGC-GA, GA + EGC-EGC, GA + EGCG-GA, GA + EGCG-EGCG, GA + EGC + EGCG-GA, GA + EGC + EGCG-EGC, GA + EGC + EGCG-EGCG in Figure (**B**) represent corresponding DRI values for single drugs in the combination of various drug groups.

**Figure 5 nutrients-14-03602-f005:**
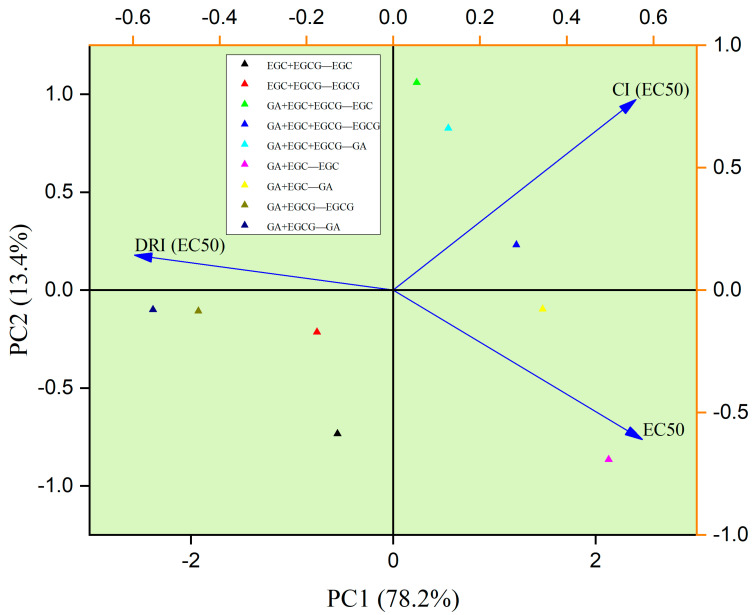
PCA among phenolic monomers of RFE.

**Figure 6 nutrients-14-03602-f006:**
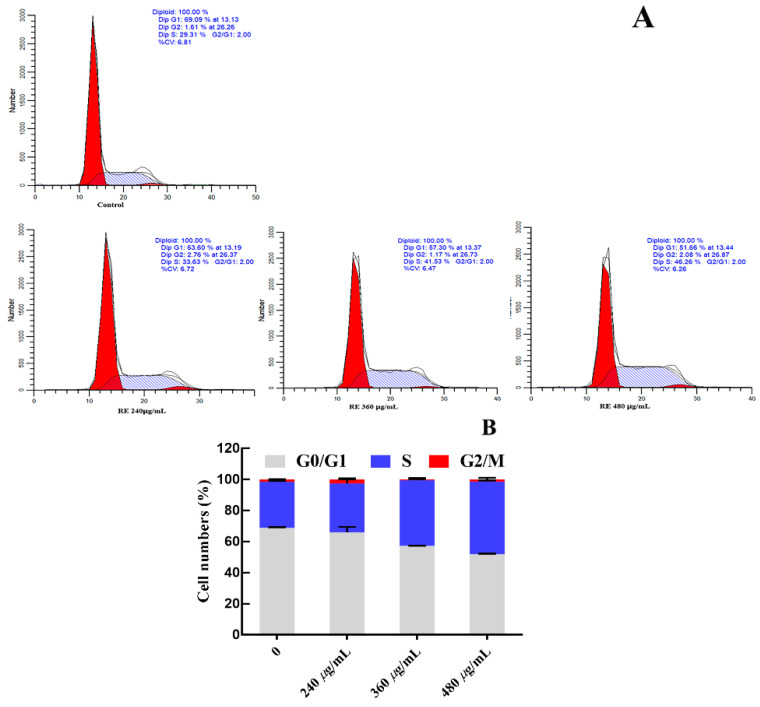
Effect of RFE on the cycle of HepG2 cells (**A**) and bar chart of effect of RFE on the cycle of HepG2 cells (**B**) (mean ± SD, *n* = 3).

**Figure 7 nutrients-14-03602-f007:**
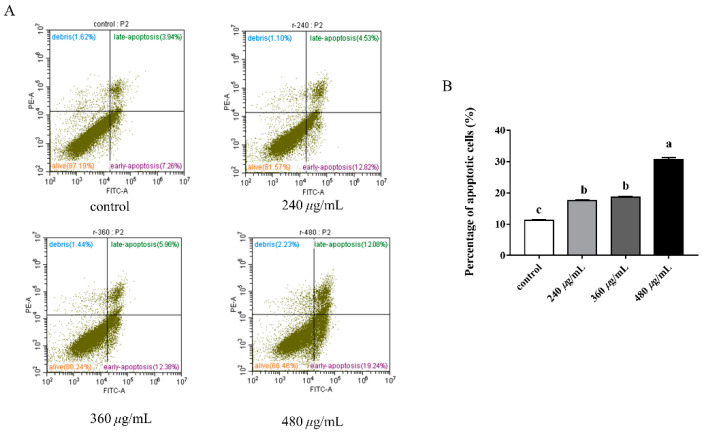
The cell annexin V/PI staining bitmap (**A**), and the effect of RFE on the apoptosis of HepG2 cells (**B**) (mean ± SD, *n* = 3). Bars with different letters differ significantly at *p* < 0.05.

**Figure 8 nutrients-14-03602-f008:**
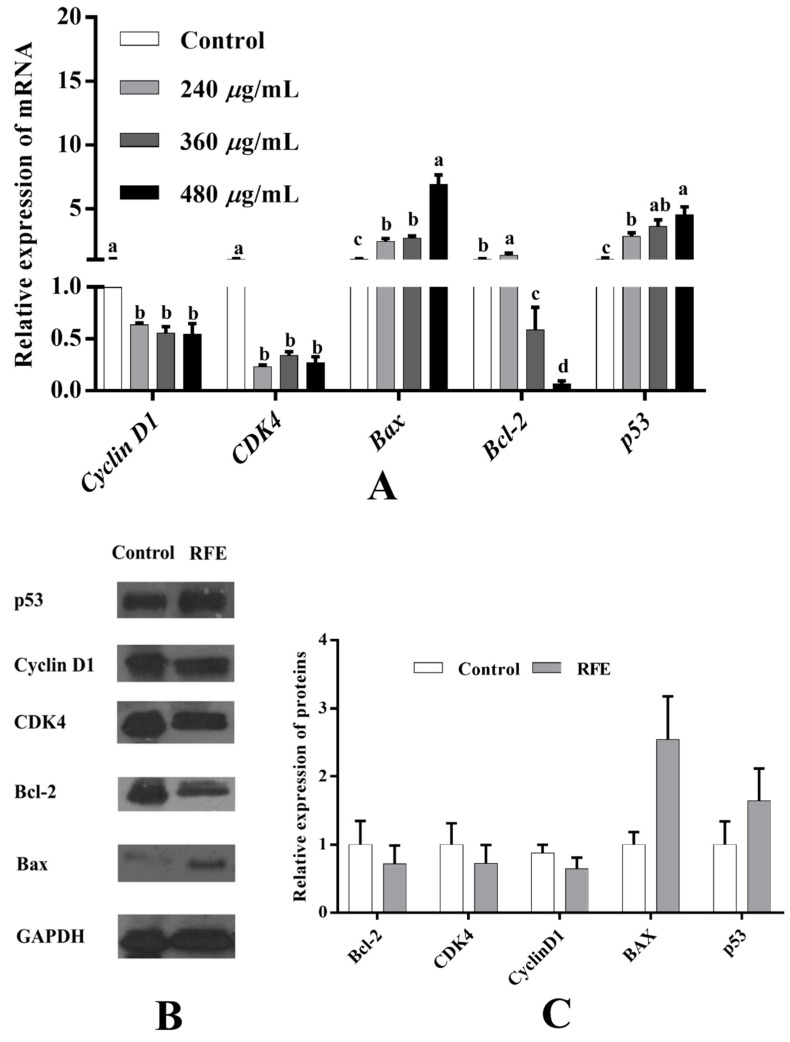
Effect of RFE on the mRNA expression of proliferation-related genes in HepG2 cells detected by RT-qPCR (**A**); protein bands of p53, cyclin D1, CDK4, Bcl-2, Bax and GAPDH (**B**); and the relative expression of proteins in control and RFE groups (**C**). Bars with different letters differ significantly at *p* < 0.05.

**Table 1 nutrients-14-03602-t001:** Phenolic profiles of *Rhodiola rosea* phenolic extract (RE).

Polyphenol	
Free polyphenol (mg GAE/g FW)	120.13 ± 2.88
Bound polyphenol (mg GAE/g FW)	2.58 ± 0.22
Flavonoids	
Free flavonoids (mg CE/g FW)	125.71 ± 6.06
Bound flavonoids (mg CE/g FW)	3.28 ± 0.24
GA (mg/100 g FW)	319.17 ± 41.18
EGC (mg/100 g FW)	267.04 ± 16.09
EGCG (mg/100 g FW)	722.10 ± 54.26
Catechin (mg/100 g FW)	35.93 ± 7.39

**Table 2 nutrients-14-03602-t002:** Volatile compounds identified in *Rhodiola rosea* by static headspace-gas chromatography–tandem mass spectrometry system.

Compound Name	Contents (ng/100 g FW)
1-Hexanol	105.51 ± 4.13
Acetic acid	514.48 ± 6.77
Benzaldehyde	68.36 ± 19.21
1-Octanol	848.14 ± 19.60
Myrtenal	90.59 ± 4.51
L-α-Terpineol	67.14 ± 4.40
2-Methyl-2-butenolide	326.18 ± 2.95
Pentanoic acid	52.72 ± 13.02
(-)-Myrtenol	1477.80 ± 54.76
Phenethyl acetate	125.98 ± 11.56
Hexanoic acid	293.22 ± 10.18
Geraniol	550.87 ± 21.27
Benzyl alcohol	416.38 ± 11.03
Phenylethyl alcohol	7869.33 ± 174.51
2-(Hydroxymethyl) but-2-enenitrile	67.66 ± 9.45
Octanoic acid	99.44 ± 7.55

**Table 3 nutrients-14-03602-t003:** V_E_ and carotenoid profiles of *Rhodiola rosea*.

Compounds (Vitamin E)	Contents (μg/100 g FW)
α-Tocopherol	338.51 ± 17.75
α-Tocotrienol	16.82 ± 0.47
β-Tocotrienol	70.47 ± 0.78
γ-Tocopherols	19.88 ± 1.32
γ-Tocotrienol	29.38 ± 3.44
δ-Tocotrienol	27.89 ± 0.53
Total	502.94 ± 17.41
Compounds (Carotenoids)	Contents (μg/100 g FW)
Lutein	3.48 ± 0.11
β-Carotene	3.95 ± 0.40
Total	7.43 ± 0.72

**Table 4 nutrients-14-03602-t004:** The EC_50_ of CAA value and CAA quality of RE (mean ± SD, *n* = 3).

Fractions	EC_50_ (µg/mL)		CAA Quality (µmol of QE/100 µmol of Polyphenols)		CAA Quality (µmol of QE/100 µmol of Flavonoids)	
	PBS Wash	No PBS Wash	PBS Wash	No PBS Wash	PBS Wash	No PBS Wash
RFE	118.49 ± 7.58	56.09 ± 0.34	10.51 ± 0.25	14.71 ± 0.35	15.51 ± 0.67	21.74 ± 0.94
RBE	10,048.57 ± 131.91	3018.09 ± 133.12	0.10 ± 0.00	0.15 ± 0.01	0.30 ± 0.01	0.44 ± 0.02
Total	-	-	10.64 ± 0.25	15.01 ± 0.36	15.67 ± 0.68	22.15 ± 0.96
EGCG	176.09 ± 9.83	52.31 ± 2.55	0.96 ± 0.04	1.64 ± 0.07	2.15 ± 0.09	3.66 ± 0.15

**Table 5 nutrients-14-03602-t005:** The EC_50_ (72 h), CC_50_ (24 h) and SI values of RFE, GA, EGC, EGCG against HepG2 cells.

Samples	EC_50_ (μg/mL)	CC_50_ (μg/mL)	SI (CC_50_/EC_50_)
RFE	276.32 ± 8.47	>1400	>2
Samples	EC50 (μM)	CC50 (μM)	SI (CC50/EC50)
GA	73.91 ± 3.23	>600	>2
EGC	107.21 ± 4.64	>600	>2
EGCG	61.66 ± 7.97	>500	>2

**Table 6 nutrients-14-03602-t006:** Inhibitory effects by combination of the main phenolic monomers in RFE (%, mean ± SD).

Concentration	Inhibitory Effects of Combinations (%)			
	GA + EGC	GA + EGCG	EGC + EGCG	GA + EGC + EGCG (2/3)
0.125 × EC_50_	18.97 ± 3.28	48.74 ± 3.18	33.81 ± 2.76	30.58 ± 5.34
0.25 × EC_50_	32.29 ± 7.09	47.97 ± 9.53	54.22 ± 5.66	37.18 ± 4.55
0.50 × EC_50_	53.57 ± 3.86	55.79 ± 2.05	57.56 ± 10.95	47.22 ± 3.89
0.75 × EC_50_	63.37 ± 9.51	72.09 ± 5.75	70.04 ± 0.92	57.69 ± 2.67
1.0 × EC_50_	68.80 ± 1.31	67.38 ± 3.14	75.63 ± 3.13	61.80 ± 0.43
1.25 × EC_50_	78.67 ± 1.60	77.40 ± 1.78	75.81 ± 4.24	67.89 ± 0.59

**Table 7 nutrients-14-03602-t007:** DRI values of single compound in HepG2 cells.

Compound	Range of DRI ^A^			
	GA + EGC	GA + EGCG	EGC + EGCG	GA + EGC + EGCG
GA	1.18 ± 0.18–4.23 ± 0.91	0.72 ± 0.12–34.48 ± 5.51	- ^B^	0.72 ± 0.09–14.26 ± 4.97
EGC	1.77 ± 0.27–3.15 ± 0.68	-	1.65 ± 0.67–7.43 ± 1.62	1.08 ± 0.13–10.63 ± 3.70
EGCG	-	1.21 ± 0.07–14.24 ± 0.95	2.00 ± 0.81–4.41 ± 0.96	1.20 ± 0.04–7.16 ± 1.03

^A^ The values of DRI in the table are 0.2 ≤ Fa ≤ 0.8. ^B^ “-” means not detected.

## Data Availability

Data will be made available on reasonable request.

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
