# Peer review of "Phytochemical Profiles, Antioxidant Activity and Antiproliferative Mechanism of Rhodiola rosea L. Phenolic Extract"

_nutrients, 2022, doi:10.3390/nu14173602_

Round 1

Reviewer 1 Report

- Related sources published after 2015 should be used more in the introduction.

- More recent references should be used in the discussion section

-Out of the 43 references used, about 18 are related to after 2010, which seems to be improved. Try to have at least 65% of the references used from the references published after 2010.

Reviewer 2 Report

Manuscript title:  Phytochemical profiles, antioxidant activity and antiprolifera-2 tive mechanism of Rhodiola phenolic extract

This study evaluated the phenolic profiles, antioxidant and antiproliferative activities.  of Rhodiola free phenolic; ainsi que the combined effect of GA, EGC and EGCG on inhibiting the proliferation of HepG2 cells. Also, a study of the effect of RFE on mitochondria pathway-related gene and protein expression was made.

In my opinion this study des clarifications, below my remarks:

Title

·         Include in brackets the species/subspecies of Ephedra alata that was used; I have included this for you in the proposed title below as an example, but you need to put the correct one as was identified by the botanist!

Abstract:

·         It is necessary to indicate the major components with % as well as anti-proliferative effect is significant or not

·         All the abbreviations should be defined at their first use in the abstract and throughout the manuscript.

Introduction

·          Include a paragraph for some studies that have tackled similar studies using other plants or similar plants

Materials and methods

·          The coordinates of the location for where the plant was collected should be provided

·         A statement on the identification of the plant by a botanist should be included

·         The statement on extraction method used needs to be cited or backed by references and this should be done for all methods

·         Why this combination of GA, EGC and EGCG? these bioactive molecules exist in several plants so I think useless. In addition, it is important to make the biological study of each single molecule; then the effect of the combination

·         A rate of plagiarism is 26% without references and citations so the rate must not exceed 20%

Reviewer 3 Report

This comprehensive research presents phytochemical profiles and biological activities of Rhodiola phenolic extract (RFE), including free and bound phenolic extracts.

The introduction section is well written; the authors described the biological activities of the main phenolics, i.e., EGCG, EGC and GA,  and volatile compounds in Rhodiola, e.g., myrtenol, geraniol, etc., based on the previous research.

The sentence in lines 62-64 needs to be corrected.

The section on Methodology is the strength of this manuscript but has some limitations. The authors mentioned that “Rhizomes of Rhodiola L. were purchased from local pharmacy in Yushu, Qinghai Province, China, and the species was identified by Prof. Shizhen Ma (Northwest Institute of Plateau Biology, Chinese Academy of Sciences, Qinghai, China) in 2019”.

Since the genus Rhodiola consists of more than 200 species, how many species have you identified? You should specify this.

Content in lines 89-91 seems to be repeated!?

Results are clearly presented in tables and figures and correspond with the main text.

This study provides many details regarding Rhodiola bioactive compounds and their potential use in clinical practice. 

Reviewer 4 Report

The manuscript entitled “Phytochemical profiles, antioxidant activity and antiproliferative mechanism of Rhodiola phenolic extract” has been evaluated and my comments are as follows,

In abstract part, line numbers (15-16) was not clear “Overall, Rhodiola phenolic extract (RE) consisted of Rhodiola free phenolic extract (RFE)”. Kindly check the following sentences.

The authors must include the details of Rhodiola plant material used for the extraction process in methodology part.

I would recommend the authors to rephrase the following sentences in introduction part [62-63]. The statement was not clear and I also suggest the authors to use punctuations appropriately.

I also recommend the authors to include how this Rhodiola polyphenol is differed from other naturally derived polyphenols. Discuss in introduction part.

Include the sentences in Introduction part. The biological potential of brown algal polyphenol dieckol has been reviewed by DK Rajan et al. (2021) and cite the following paper. https://doi.org/10.1016/j.biopha.2021.111988

The error bars in Figure. 2 is diminished and I also recommend the authors to replace the existing figure file with new one. Especially, the graphical representation lacks of error bars in ORAC (c) value.

Why the authors have directly done HPLC analysis alone for confirming the phenolic profiles in Rhodiola? It is mandatory to do functional group analysis for Rhodiola based compounds.

The fluorescent microscopy images representing the cytotoxic potential of RFE will add more strength to the paper. Hence, I would recommend the authors to include the microscopic images of RFE treated HepG2 cells.

The concentrations (360 µg/mL and 480 µg/mL) shown in Figure. 7 looks similar.

In Figure -8- I haven’t find any band patterns for Bax. Kindly clarify or give your explanations.

Overall, this research manuscript contains some linguistic issues. Hence, I recommend the authors to edit the manuscript with native English speaker before submitting the revised version.

Round 2

Reviewer 4 Report

The authors successfully addressed the reviewer comments or concerns. If possible, the authors can add the fluorescent microscopy images representing the cytotoxic potential of RFE in the manuscript.
